# Fifteen-Years Follow-Up in a Cohort of Children with Functional Gastrointestinal Disorders: Prevalence and Risk Factors to Develop Neuropsychiatric Disorders and Other Comorbidities

**DOI:** 10.3390/children8100838

**Published:** 2021-09-24

**Authors:** Chiara Zanchi, Stefano Pintaldi, Grazia Di Leo, Luca Ronfani, Giulia Zamagni, Marialuisa Viel, Egidio Barbi, Giorgio Cozzi

**Affiliations:** 1Department of Pediatrics, Institute for Maternal and Child Health “IRCCS Burlo Garofolo”, Via dell’Istria 65, 34137 Trieste, Italy; grazia.dileo@burlo.trieste.it (G.D.L.); egidio.barbi@burlo.trieste.it (E.B.); giorgio.cozzi@burlo.trieste.it (G.C.); 2Department of Medicine, Surgery and Health Sciences, University of Trieste, 34127 Trieste, Italy; pintaldistefano@gmail.com (S.P.); marialuisaviel94@gmail.com (M.V.); 3Epidemiology and Biostatistics Unit, Institute for Maternal and Child Health “IRCCS Burlo Garofolo”, 34137 Trieste, Italy; luca.ronfani@burlo.trieste.it (L.R.); giulia.zamagni@burlo.trieste.it (G.Z.)

**Keywords:** functional gastrointestinal disorders, headache, migraine and chronic intestinal diseases

## Abstract

Background: Functional gastrointestinal disorders (FGIDs) are chronic and recurrent disorders, which affect up to 23% of children and adolescents and represent 50% of gastroenterological accesses. The association between FGIDs diagnosed at paediatric age and the onset of migraine or headache and neuropsychiatric diseases in adolescence and adulthood is widely reported in the literature. However, there is still limited knowledge about the long-term prognosis and risk factors for neuropsychiatric pathologies and other comorbidities. Aim: The aim is to assess the prevalence and persistence of FGIDs as well as the occurrence of migraine or headache and neuropsychiatric disorders in a cohort of patients diagnosed with FGIDs 15 years ago compared with a control group of peers. Materials and methods: We enrolled a group of patients diagnosed with FGIDs at paediatric age, at least 10 years ago (FGIDs group, *n* = 79; median age 23), and control subjects (control group, *n* = 201; median age 23). In both groups, an online questionnaire created explicitly for the study was submitted in order to investigate the presence of chronic intestinal diseases, migraine, headache or neuropsychiatric disorders. Results: 45.6% (36 out of 79) of patients previously diagnosed with FGIDs still suffer from FGIDs versus 12% (24 out of 201) of healthy controls (*p* < 0.0001). The prevalence of chronic organic gastrointestinal disorders was comparable in the two groups (2.5% in FGIDs group versus 1% in healthy group, *p* = 0.3). Thirty-three percent (26 out of 79) of FGIDs patients reported headache or migraine versus 13% (26 out of 201) of healthy peers (*p* < 0.001). No differences were found regarding the prevalence of anxiety and depression. Conclusion: The outcome at 15 years of FGIDs was characterized by a high prevalence of persisting functional symptoms along with a significant incidence of headaches and migraines. Abbreviation: FGIDs: Functional gastrointestinal disorders; IBS: Inflammatory Bowel Syndrome.

## 1. Introduction

Functional gastrointestinal disorder (FGID) is a term used to describe a group of different chronic and recurrent gastroenterological clinical pictures that cannot be explained by underlying structural or tissue abnormalities [1]. FGIDs have an essential impact on the quality of life related to health, private life, education and work. Only a few studies have evaluated the natural clinical history of FIGDs and their correlation with other comorbidities during adulthood age.

This study aims to verify the persistence of FGIDs, the prevalence of organic chronic gastrointestinal disorders (Crohn’s disease, ulcerative colitis and coeliac disease), headache, depression and anxiety disorders in a group of young adults who had received FGIDs diagnosis during childhood and to compare them with a control group of the general population.

## 2. Methods

Our study is a historic cohort follow-up study. It was carried out at the outpatient clinic of the pediatric gastroenterology and the pediatric ward of the Institute for Maternal and Child Health IRCCS Burlo Garofolo of Trieste, Italy. The study project was approved by the Institutional Review Board of the institute.

The medical records of all children who received a diagnosis of FGIDs between 2002 and 2010 were retrospectively reviewed. The inclusion criteria were as follows: age at diagnosis from 2 years to 17 years; and FGIDs diagnosis according to the Rome III criteria. Patients with the presence of any organic gastrointestinal disease at the time of the FGIDs diagnosis were excluded.

The cohort of eligible patients was contacted by email, and then the purpose of the study was explained with a phone call, and informed consent was obtained. The subjects enrolled in the study received an online questionnaire.

The questionnaire was specifically developed for the study to investigate the following items: having received a diagnosis of FGIDs or of any organic gastrointestinal disease, neuropsychiatric disorders, headache and migraine from a general practitioner or a medical specialist.

In order to compare the answers of the questionnaire of the study cohort, a group of healthy subjects, matched for age, gender and living in the same geographic area, were recruited and formed the study control group. These subjects were contacted via a social platform commonly used among university students.

The primary study outcome was the prevalence of FGIDs in adulthood in the historical cohort of children who have already received a diagnosis of FGIDs during childhood compared to the healthy subjects.

The secondary outcomes were the prevalence of organic gastrointestinal diseases, neuropsychiatric disorders and headache/migraine in both groups.

## 3. Statistical Analysis

Categorical data were presented as absolute frequencies and percentages, while continuous data were reported as mean and standard deviation.

The differences between the two groups of comparison (patients with previous FGIDs and healthy controls) were evaluated by using the Fisher’s exact test for categorical variables and the Student’s t-test for continuous variables.

Bivariate and multivariate logistic regression analyses were conducted in order to identify the risk factors associated with neuropsychiatric disorders (anxiety and depression) and migraine/cephalalgia.

Statistical significance was set at *p* < 0.05.

The data were analyzed with StataCorp. 2019. Stata Statistical Software: Release 16. College Station, TX, USA: StataCorp LLC.

## 4. Results

Of the 160 individuals with FGIDs who met the inclusion criteria, 79 (49%) accepted participation and answered the questionnaire. The time elapsed between the diagnosis and the investigation time was, on average, 15 years, ranging from 10 to 18 years. The clinical and demographic characteristics of the cohort are summarized in Table 1. The age of the patients at the first FGIDs diagnosis was 9 years on average, and the age of the same patients at the time of questionnaire administration was 23 years on average. The control group included 201 subjects from the general population matched by age and sex to balance the study group (see Table 1).

The spectrum of FGIDs diagnosed at pediatric age and at follow up in the patients’ group showed a prevalence of functional abdominal pain (59%) followed by functional constipation (20.3%), irritable bowel syndrome (IBS) (11.4%), cyclic vomiting (11.4%) and functional dyspepsia (6.6%). At the time of investigation, 36 out of 79 of the cases (45.6%) still had at least one FGID (F 63.9%, median age 24 ± 5 years), with an equal distribution across the disorders: IBS (20.2%), functional dyspepsia (20.2%), functional abdominal pain (20.2%) and functional constipation (19%). Twenty-six out of seventy-nine individuals (33%), suffered from migraine-or-headache. Fifteen out of seventy-nine cases (19%) reported anxiety disorder, and 5 out of 79 subjects (6.3%) presented depression disorder. No schizophrenia diagnosis has occurred (see Table 1). Stratifying data according to gender revealed a clear predominance of neuropsychiatric disorders in females, since anxiety and depression occurred in 32% of females in contrast to 10% of males (OR 4.2, CI 95% 1.23–14.30 and *p*-value = 0.02, see Table 2); moreover, a migraine and headache also occurred in 45% of females and 20.5% of male (OR 3.2, CI 95% 1.17–8.50 and *p*-value = 0.02, see Table 3).

The cross-sectional univariate analysis within the study group showed that the persistence of FGIDs at the time of investigation was a risk factor for developing headache and migraine, since 44% (16 out of 36) of subjects with FGIDs’ persistence received diagnosis of headache or migraine (OR 2.64; CI 95% 1.01–6.93, see Table 3). Nineteen out of twenty-six (73.1%) patients who suffer of headache and migraine had received diagnosis of functional abdominal pain in pediatric age: 6 out of 26 (23.1%) had received diagnosis of functional constipation and 2 out of 26 (7.7%) had received diagnosis of IBS and cyclic vomiting. Moreover, 16 out of 26 (61.5%) patients who suffer of headache and migraine showed persistence of FGIDs at the time of follow-up.

The case-control analysis showed a statistically significant higher prevalence of headache-or-migraine, with 33% in the study group versus 12.9% in the control group (*p* value < 0.001, see Table 1). No association was found between the pediatric history of FGIDs and the diagnosis of neuropsychiatric disorders such as anxiety, depression and schizophrenia (see Table 1).

Out of the cases, 2.5 % of cases (2 out of 79) within the study group developed coeliac disease, without any statistically significant difference with the control group. Both had tested negative for coeliac disease at the time of diagnosis of FGID in pediatric age. No other abdominal organic disease occurred either in the study and the control groups.

## 5. Discussion

The main finding of this study is the significant persistence of FGIDs in patients diagnosed during childhood, without organic chronic gastrointestinal disorders. As expected, the prevalence of all types of FGIDs in adulthood age was significantly higher in patients who had been diagnosed at paediatric age than in the control group.

Despite the high prevalence of FGIDs and their known comorbidities, there is still limited evidence in the literature of the long-term outcomes relative to children who received this diagnosis.

FGIDs have a pooled prevalence of 8–25% of school-age youth from 8 to 18 years and are associated with school absences, poor academic performance and difficulties with peer relationships [2,3]. The diagnosis relies on symptom-based criteria, the current Roma IV criteria and exclusion in the cost-effective practice of other specific conditions with similar clinical presentation [4,5].

The pathophysiology of FGID is complex and involves a bidirectional dysregulation of the gut–brain interaction (named «the gut-brain axis») visceral hypersensitivity, abnormal gastrointestinal motility and decreased threshold for pain in response to changes in intraluminal pressure not determined by structural or biochemical anomalies [4,5]. Both adults and children with FGID may have abnormal bowel reactivity with respect to physiologic stimuli (meal, gut distension and hormonal changes), noxious stressful stimuli (inflammatory processes) or stressful psychological stimuli (parental separation and anxiety) [2,3,4,5].

Psychological and neuropsychiatric comorbidities are frequently reported [2,3,4]. The possible relationship between FGIDs and neuropsychiatric diseases is supported by the evidence that there are several areas of abnormal brain activity in patients with FGIDs associated with visceral hypersensitivity, anxiety and depression [4,6,7]. Recent epidemiological studies suggest that, in 50% of cases, FGIDs begin with psychological distress, followed later by gastrointestinal symptoms; in the remaining 50% of cases, gut dysfunction occurs first, and psychological distress follows later [3,8,9]. The association between some types of FGIDs and migraine in children and adolescent has also been described, hypothesizing similar pathogenetic mechanisms [1]. Hence, the hypothesis is that gastrointestinal and psychological manifestations are an integral part of the same process driving the disease [8,9,10,11].

The results of our study are in line with those in the literature that defines FGIDs as “chronic conditions”, characterized by fluctuation and persistence of symptoms during a lifetime with frequent shifts towards a different type of FGIDs [4,12,13,14,15]. Previous longitudinal studies showed an association between the history of functional abdominal pain in childhood and the presence of FGIDs in adolescence and young adulthood, with a prevalence between 35 and 41% [16]. To our knowledge, our study is the first to evaluate the evolution of all subtypes of FGIDs in childhood. It considers not only recurrent abdominal pain but also describes the evolution of the different forms in the same cohort. Overall, these data showed an equal distribution of IBS, dyspepsia, functional constipation and recurrent abdominal pain in adulthood, while cyclic vomiting seemed to disappear (see Figure 1). At pediatric age, functional abdominal pain is the most common subtype of FGIDs. During growth, its prevalence decreases, and there is no prevalent subtype of FGIDs later in adulthood. Dyspepsia, IBS and functional constipation are the more common FGIDs in adults, and symptom overlap is frequently observed, as was observed in our study, so that two or more type of FGIDs can coexist [4]. The literature prevalence of cyclic vomiting syndrome is the lowest between FGIDs both in adults than in children (respectively, 0.2–1% and 1%), while recurrent abdominal pain affects a mean of 35–38% of elementary school children [4,16].

This study also highlights a high risk of headaches and migraines in adults who had received FGID diagnosis at paediatric age, compared with control groups (33% vs. 13%, *p* < 0.001), and its incidence was significantly higher in female patients than in male patients. Functional abdominal pain was the pediatric FGID related to the high risk of developing migraine and headache in adulthood. Moreover, the persistence of FGIDs over time was related to the possibility of suffering migraines and headaches in adulthood (61.5% of patients with migraine or headache at the follow-up presented persistent FGIDs). In the literature, a limited number of studies have addressed the concomitant association between headache, migraine and FGIDs.

Furthermore, there is no previous study investigating the risk of developing headache and migraine in subjects who had received a FGID diagnosis at paediatric age. A recent case-control study showed a positive association between migraine in children and functional dyspepsia, IBS and abdominal migraine (32% of pediatric patients with migraine also reported a history of FGIDs versus 18% of control, *p* < 0.00001), suggesting that FGIDs and migraine can coexist in the same paediatric patient. The authors speculate that the gut-brain axis not only plays vital role in the association between FGIDs and migraine and headache, but multiple inflammatory and vasoactive substances could also be involved in the pathogenesis of both disorders [1,17,18,19,20,21].

The strict relationship between FGIDs and migraine or headache was also proven by interesting reports that showed the efficacy of intranasal sumatriptan, commonly used to treat migraine, in pediatric cases of abdominal pain-related FGID [22]. This occurrence should not be surprising, since sumatriptan is a serotonin (5-hydroxytryptamine) 1B/1D agonist, and serotonergic dysfunction is involved in both migraines and FGIDs. Moreover, cyclic vomiting syndrome, abdominal migraine, benign paroxysmal vertigo and benign paroxysmal torticollis are considered pediatric migraine forms [23].

A recent Korean prospective study showed that 70% of the adult hospital-wide population with migraine met the Roma III criteria for concurrent FGID, and 40% met the criteria for more than one FGID. These authors demonstrated a clear link between coexistent FGID symptoms, psychological comorbidities and the existence of a “load effect”, as anxiety/depression scores worsened with increasing numbers of FGID symptoms [24].

The third finding of our study is the absence of a correlation between the pediatric history of FGIDs and the presence of neuropsychiatric disorders, such as anxiety and depression in adulthood. These data are inconsistent with the current literature, which instead demonstrates a higher prevalence of anxiety and depression in patients who suffer or have suffered from FGIDs [8,9,10,11]. We believe that these data should be contextualized in the current historical reality in which the COVID-19 pandemic may have increased the prevalence of neuropsychiatric symptoms such as anxiety and depression in the general population. Recent literature suggests that the prevalence of depression and anxiety symptoms was, respectively, more than 3-fold higher and nearly 2-fold higher during COVID-19 compared with the situation before the pandemic, especially in the female sex and university students [25,26,27].

This study has some weakness: It is a retrospective study with a significant loss to follow-up; we relayed on a reported medical diagnosis, and we did not use a validated questionnaire to assess the prevalence of neuropsychiatric disease, migraine, headache and chronic intestinal disease. We also excluded people who had received a paediatric diagnosis of FGID from the control group.

The point of strength of this study is its cross-sectional nature with a long-term follow-up.

## 6. Conclusions

The outcome at an average of 15 years of FGID was characterized by a high prevalence of persisting functional symptoms with a significant incidence of migraines and headaches without increased risk of anxiety or depression.

## Figures and Tables

**Figure 1 children-08-00838-f001:**
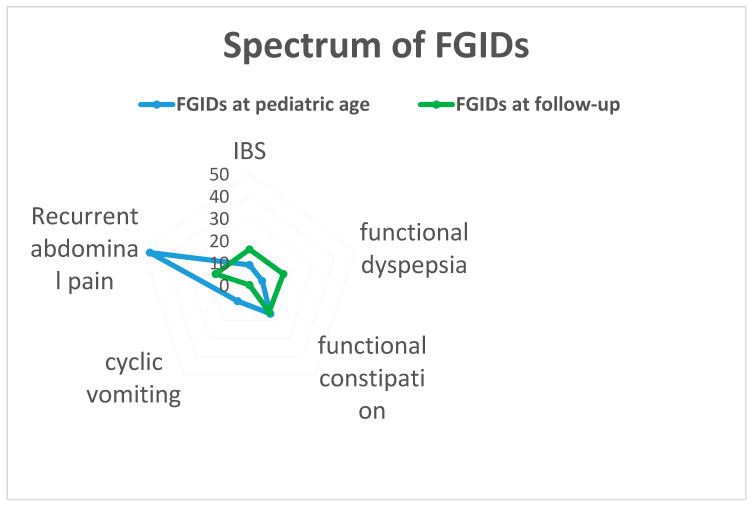
The figure shows the distribution of all subtypes of FGIDs during childhood (blue line) and adulthood (green line), representing their evolution over time. The most frequent FGID type in childhood is recurrent abdominal pain. Otherwise, in the young adult population dyspepsia, IBS, functional constipation and recurrent abdominal pain are equally represented, while cyclic vomiting seems to disappear.

**Table 1 children-08-00838-t001:** Descriptive and comparative table of study and control groups.

	Control Group *n* = 201	Case Group *n* = 79	*p*-Value
Current Age, mean (SD)	23.0 (2.7)	23.2 (5.4)	0.833
Sex, *n* (%)			
Males	78 (38.8)	39 (49.4)	0.138
Females	123 (61.2)	40 (50.6)	
FGIDs at pediatric age, *n* (%)			
IBS		9 (11)	
Functional dyspepsia		6 (8)	
Functional constipation		16 (20)	
Cyclic vomiting		9 (11)	
Functional abdominal pain		47 (59)	
FGIDs at follow-up, *n* (%)			
c	14 (7.0)	16 (20.2)	0.002
Functional dyspepsia	4 (2.0)	16 (20.2)	<0.001
Functional constipation	5 (2.5)	15 (19.0)	<0.001
Cyclic vomiting	1 (0.5)	0	1.000
Functional abdominal pain	10 (5.0)	16 (20.2)	<0.001
Organic disease, *n* (%)			
Coeliac disease	2 (1.0)	2 (2.5)	0.316
Migraine or headache, *n* (%)	26 (12.9)	26 (32.9)	<0.001
Psychiatric disorders, *n* (%)			
Anxiety	26 (12.9)	15 (19.0)	0.195
Depression	13 (6.5)	5 (6.3)	1.000

FGIDs: Functional Gastrointestinal Disorders; IBS: Irritable Bowel Disease.

**Table 2 children-08-00838-t002:** Risk factors for anxiety or depression.

			Univariate			Multivariate		
Anxiety or Depression	Events	*n*	OR	95% CI	*p*-Value	OR	95% CI	*p*-Value
Current age	17	79	1.07	(0.96; 1.18)	0.215	1.02	(0.91; 1.13)	0.738
Sex								
Male	4	39						
Female	13	40	4.21	(1.23; 14.38)	0.022	3.96	(1.02; 15.36)	** *0.046* **
Any FGIDs at current age								
No	7	43						
Yes	10	36	1.98	(0.67; 5.88)	0.220	1.69	(0.50; 5.76)	0.402

**Table 3 children-08-00838-t003:** Risk factors for migraine or headache.

			Univariate			Multivariate		
Migraine or Headache	Events	*n*	OR	95% CI	*p*-Value	OR	95% CI	*p*-Value
Current age	26	79	1.00	(0.92; 1.09)	0.998	0.95	(0.86; 1.05)	0.311
Sex								
Male	8	39						
Female	18	40	3.17	(1.17; 8.58)	0.023	2.47	(0.83; 7.34)	0.104
Any FGIDs at current age								
No	10	43						
Yes	16	36	2.64	(1.01; 6.93)	0.049	2.11	(0.73; 6.12)	0.170

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
