# Peer review of "Fifteen-Years Follow-Up in a Cohort of Children with Functional Gastrointestinal Disorders: Prevalence and Risk Factors to Develop Neuropsychiatric Disorders and Other Comorbidities"

_children, 2021, doi:10.3390/children8100838_

Round 1

Reviewer 1 Report

This study showed that FGID in children is a risk factor associated with migraine or headache by fifteen-year follow-up in a cohort of children. The point of view and the results are interesting. There are several issues that should be addressed.

  1. The frequent FGID subtypes are functional abdominal pain and cyclic vomiting in childhood whereas IBS and functional dyspepsia in adulthood. The point on these differences should be described in discussion.

  1. Is there a statistical significance in the number of male and female between control group and case group? P-value should be shown in Table 1.

  1. page 3, line 115; Inflammatory → Irritable

  1. page 3, line 126; see table 2 → see table 3

Reviewer 2 Report

Dear Author
Thank you for submitting this work.
The weight of functional diseases in childhood is sometimes forgotten and
the expected evolution of these situations is often overlooked.

Despite the interest of this work, I think it could be improved. I consider the conclusions to be difficult to assess. For example, the high persistence of symptoms is mentioned, however,
55% of patients no longer have GI functional pathology
It is also not clear the evolution of associated symptoms, for example,
in headache and migraine, there is no reference to which they have been
investigated in patients who, at the current age, no longer have FGID (table 2 and 3)
Enclosed text with comments
Many regards
